# The Effects of Iron on In Silico Simulated Abiotic Reaction Networks

**DOI:** 10.3390/molecules27248870

**Published:** 2022-12-13

**Authors:** Sahil Rajiv Shahi, H. James Cleaves

**Affiliations:** 1Blue Marble Space Institute of Science, Seattle, WA 98104, USA; 2Sardar Vallabhbhai National Institute of Technology, Surat 395007, India; 3Earth-Life Science Institute, Tokyo Institute of Technology, Tokyo 152-8550, Japan; 4Earth and Planets Lab, The Carnegie Institution for Science, Washington, DC 20015, USA

**Keywords:** iron chemistry, prebiotic chemistry, origins of life, combinatorial chemistry, chemical reaction networks, iron-sulfur world, formose reaction, Maillard reaction, pyruvic acid, glucose

## Abstract

Iron is one of the most abundant elements in the Universe and Earth’s surfaces, and undergoes a redox change of approximately 0.77 mV in changing between its +2 and +3 states. Many contemporary terrestrial organisms are deeply connected to inorganic geochemistry via exploitation of this redox change, and iron redox reactions and catalysis are known to cause significant changes in the course of complex abiotic reactions. These observations point to the question of whether iron may have steered prebiotic chemistry during the emergence of life. Using kinetically naive in silico reaction modeling we explored the potential effects of iron ions on complex reaction networks of prebiotic interest, namely the formose reaction, the complexifying degradation reaction of pyruvic acid in water, glucose degradation, and the Maillard reaction. We find that iron ions produce significant changes in the connectivity of various known diversity-generating reaction networks of proposed prebiotic significance, generally significantly diversifying novel molecular products by ~20%, but also adding the potential for kinetic effects that could allow iron to steer prebiotic chemistry in marked ways.

## 1. Introduction

Iron (atomic symbol Fe) is the most abundant element overall by mass on Earth [1]), and is also one of the most abundant elements in Earth’s crust (~6.3%, [2]). Iron is also a central player in bioinorganic chemistry, being involved in a large number of intracellular and environmental biochemical redox transformations [2,3] and playing an important though under-appreciated role in modern anthropic chemistry [4].

The environmentally prevalent oxidation state of iron is governed by the redox environment (e.g., [5]). While it is suspected that iron in Earth’s core is largely present in the Fe^0^ or metallic state, the iron in Earth’s upper mantle and crust is present in either the ferrous (Fe (II) or Fe^2+^) or ferric (Fe (III) or Fe^3+^) state. Converting Fe (II) to Fe (III) requires approximately −770.1 mV [6] at standard temperature and pressure. This energy difference is approximately enough to compensate for various functional group changes, for example, the reduction/oxidation of ketones (which ranges from 60–350 mV, with more typical values of species of biochemical interest falling in the 200 mV range [7]). However, such energetic couplings are rarely direct in biochemistry and often mediated by cofactor or membrane potential-based processes, and their redox potentials can be highly variable (e.g., [8]).

Given iron’s environmental ubiquity and importance in contemporary biochemistry, it is also widely believed that iron catalysis may have been important for life’s emergence (e.g., [9]). Different models suggest various roles for iron in life’s origin. For example, the Iron-Sulfur World model [10] suggests a role for both iron and sulfur as primitive energy sources for the emergence of life. Weber [11] demonstrated the catalysis of hexose and hydroxy acid synthesis from glyceraldehyde in the presence of aqueous Fe^3+^. Barge et al. [9] reported the reduction of pyruvate to lactate and its reductive amination to alanine in the presence of iron oxyhydroxide. Muchowska et al. [12] have reported an iron-catalyzed reaction network with pyruvate and glyoxylate that produces many intermediates and reactions of core biological pathways, such as the Krebs cycle. Furthermore, studies have found marked effects of iron (II) and (III) on the progress and outcome of complex reactions such as the Maillard reaction [13], which is a generic term for reactions of amines with reducing sugars which tend to generate complex product mixtures.

While the idea of iron-mediated prebiotic chemistry has been frequently discussed for over 30 years, few studies have explored the possible effects of iron on complex prebiotic reaction networks. There are two major impediments to understanding the impacts of iron on prebiotic chemistry. First, kinetic data from carefully controlled studies of aqueous organic reactivity in the presence of dissolved iron species are limited in the chemical literature. Chemical transformations that are possible in the absence of iron may also be catalyzed by iron species; similarly, reactions that are impossible in the absence of iron may become possible when the iron is present. Second, prebiotic chemistry includes a number of reactions that generate high chemical diversity seeded by simple likely prebiotic reactants, including HCN oligomerization [14], the formose reaction [15] and Miller-Urey type reactions [16], among others. Such reactions may produce hundreds to millions of distinct product types (e.g., [17]). Complex chemical reaction networks, such as those which gave rise to the complex organic suites observed in carbonaceous meteorites (e.g., [18]), are often assumed to have been important for the origins of life [19].

Recently, it has been shown that large complex reaction networks (CRNs) can be accurately and efficiently modeled using graph theory-based reaction network expansion techniques [20] to produce complex reaction network representations (CRNRs). These CRNRs are computed using reaction rules that are in silico representations of known real-world chemical reaction mechanisms. This reaction network expansion technique allows rapid CRNR modification by introducing new reaction rules. While such techniques are “kinetically naive”, e.g., they are not able to predict kinetics, and are practically computationally limited by the mass and number of compounds/reactions they can model, they can nevertheless accurately reproduce complex reported product suites, match high-resolution mass spectra of such reaction mixtures, and reproduce compositional trends in extended Van Krevelen and Kendrick plots from experimental data [20].

We examine here the effects on prebiotic CRNRs of adding reaction rules involving iron ions, which can allow for easy comparison and identification of likely changes in network topology caused by their inclusion or omission. The workflow used here (see Figure 1) explores how the inclusion of known organic transformation reaction mechanisms mediated by iron may affect the structure of CRNRs of possible prebiotic interest and, thus, the potential effects of iron catalysis at the origins of life.

## 2. Methods

### 2.1. Reaction Network Generation

We used the MØD software package [21] to elaborate the CRNRs investigated here. MØD is a chemical reaction network generator that uses reaction mechanisms as rules. These rules are used to generate reaction networks of input graphs (reactants) which are transformed into output graphs (products). Various constraints can be imposed on the rule set to ensure that the generated reaction network is specific and not a generalized combinatorial set of all possible graphs, i.e., it accurately represents real-world chemistry. We note that previous applications of these methods, even using 200 AMU expansion cutoffs, manage to faithfully reproduce diverse product suites detected in real world chemistry, match high-resolution mass spectra of real reactions, and faithfully predict trends in real chemistry-derived Van Krevelen and Kendrick plots [20], attesting to these methods’ ability to predict real world chemistry. We briefly review the use of these methods here, more complete information can be found in [20]. This pipeline is open-source, written mostly in Python, and can be accessed along with relevant documentation at https://github.com/Reaction-Space-Explorer/reac-space-exp (accessed on 10 July 2021).

In MØD reaction mechanisms are compiled in GML format. Molecules are represented as graphs, where edges represent bonds and vertices represent atoms. During a reaction, removal and addition of edges within a graph (representing a molecule), represents the making and breaking of chemical bonds as occurs in a chemical reaction.

A typical MØD reaction representation is shown in Figure 2. Here, the bonds in the L (for “left”) graph are those to be broken and those in the R (for “right”) graph are those to be created, and K (for the German “Kontext”) represents a common context graph of bonds that remain unaltered in the reaction. The molecule(s) represented in L are the reactant(s) and the molecule(s) resulting from the graph transformation in R are the product(s) of the reaction.

MØD operates by employing a library of user-provided reaction rules which are applied to a set of initial reactants, giving rise to an initial set of products. In constructing reaction networks, a starting set of compounds is loaded into the program, then each is compared against the set of allowed reaction rules to see if a reaction is possible. If a reaction is possible, the reactant, reaction mechanism and products are stored into memory. If a reaction is not possible, the program proceeds to the next compound and makes the same determination. Here, the input reactants are referred to as the “generation zero” (or G0) molecules or set, while the initial set of products are referred to as “generation one” (or G1). After the computation of the first set reactions and their products, the G0 and G1 molecules are combined and a second round of reactions from the library is computed, yielding a G2 product set, and this process iterated. This process can in principle be continued for any number of iterations (or generations) decided by the user, causing the network to grow at each step. In practice, the networks may grow to be so large that their computation becomes cumbersome. After completion of all desired (or possible) reaction iterations, the entire network can be saved in a format that can be analyzed using other computational tools and further analyses such as comparison with experimental data, computing molecular descriptors, and evaluation for particular reaction sequences or cyclic reaction motifs.

To control the combinatorial explosion of reaction networks, an upper mass limit of 200 units was imposed here. This number is somewhat arbitrary but practical, as it allows for multiple reaction iterations to be computed starting from relatively low molecular weight reactants, for example species of prebiotic interest including HCN, HCHO, NH_3_, simple sugars, amino acids, etc., using modest computational resources. For illustration, a comparison of the relative computation times for the generations of the alkaline degradation of glucose are reported in [20], and range from a fraction of a second for G1 to several days for G5, with computation time typically increasing exponentially with each iteration. This approach also allows for fairly exhaustive exploration of the small molecule chemical space of these reactions. Although applying such molecular weight limits does not thus allow for the discovery of long polymers, it does allow for fairly exhaustive exploration of monomer synthesis pathways. Modeling the elongation of monomers into polymers can be accomplished using these methods, but doing so alongside the exploration of the concomitant small molecule chemistry would be computationally inefficient.

In this study, rules were applied to several sets of input “reactant” graphs over a number of iterations to produce two types of networks for each input reactant set: one including iron-dependent reaction rules and another lacking them. This parallel reaction network expansion allows for a facile comparison of the effects of the addition of iron species on the network.

### 2.2. Reaction Mechanism Selection

To model the effects of iron catalysis on complex prebiotic organic reaction networks, we first systematically searched the chemical literature using Scifinder (https://scifinder.cas.org (accessed on 15 July 2021); [22]) for reactions that satisfied the following two criteria (guided in part by the work of [11,12,23,24]): 1. involved aqueous Fe^3+^ and/or Fe^2+^ ions and 2. were conducted in anoxic water under “moderate” temperature and pH values, i.e., between 25 °C to 120 °C and pH between 2 and 12 (e.g., [25]). Note that while organisms have been found living at lower temperatures, there are few examples of studies of aqueous iron chemistry at lower temperatures in the literature; thus, this reaction search is defined by reactions detectable over practical laboratory timescales. Presumably, reactions that occur at higher temperatures will also proceed, albeit more slowly, at lower ones. We did not include reactions promoted by Fenton reagents (e.g., peroxides and Fe^2+^) in water to avoid oxygen-dependent and radical-based chemistry or reactions promoted by iron-sulfur and other multi-elemental complexes, though such chemistry is of interest for future studies. More general and inclusive reaction networks can, of course, be constructed and evaluated using these methods.

We classified reactions into two categories: 1. exclusively iron-catalyzed reactions and 2. reactions that can also occur in the absence of iron, e.g., for which iron species merely play the role of catalyst (see Figure 2). For example, the oxidation of a hydroxyl group by Fe^3+^ would be classified in the first category as an iron-dependent reaction that specifically involves the exchange of electrons between iron and carbon, while reactions such as aldol condensations which can be catalyzed by iron species would be classified into the second category. 

Further examples of reactions as encoded in MØD illustrating these differences are shown in Figure 3. Note that the reaction mechanisms induced by iron ions were coded separately to differentiate between the distribution of products produced in iron-free and iron-containing reaction mechanisms.

It should be noted that these methods do not keep track of the balance of atoms in the system, but rather assume there is sufficient mass in the system for a reaction to occur if the mechanistic conditions for its occurrence are satisfied, namely that suitable reactants exist in the network. For example, there need not be a balancing mechanism to regenerate Fe^2+^ from Fe^3+^ or vice versa, although numerous reactions may in practice accomplish this.

### 2.3. Data Visualization

The data generated in MØD was output in comma-delimited text files and processed using Pandas [26]—a python data analysis library, matplotlib and MS Excel for generating plots, and Gephi [27] to produce network plots and network metric calculations to quantify the differences between iron-containing and iron-free CRNRs. Specifically, Gephi allows for facile computation of network metrics including node degree (the number of edges (or reactions) connected to each node (or molecule)), node in-degree and out-degree (measuring the number of times a node/molecule occurs in the network as a reaction product or reactant, respectively), among other metrics.

### 2.4. Reaction Network Exploration

We specifically explored four complex reaction network types to understand the potential effect of iron ions on CRNs: the formose reaction (e.g., [15]), the reaction of concentrated pyruvic acid (which degrades and self-condenses in water to give a complex product suite, e.g., [28]), glucose degradation (e.g., [20,29]) and the Maillard reaction (e.g., [30]), which is a generic name given to extreme diversity generating reactions derived from amines and reducing sugars. Notably, with the exception of the Maillard reaction, these reactions exclusively involve CHO chemistry since the addition of heteroatoms such as S, and particularly N, requires resolving more complex issues such as the treatment of tautomerism in the construction of CRNRs. 

The four reactions modeled were seeded with the following input reactants:
Formose Reaction NetworkHCHO + H_2_OHCHO + H_2_O + Fe^2+^ + Fe^3+^ + H^+^ + OH^−^Concentrated Pyruvic Acid NetworkPyruvic Acid + H_2_OPyruvic Acid + H_2_O + Fe^2+^ + Fe^3+^ + H^+^ + OH^−^Glucose Degradation Reaction NetworkGlucose + H_2_OGlucose + H_2_O + Fe^2+^ + Fe^3+^ + H^+^ + OH^−^Glucose-Glycine Maillard Reaction NetworkGlycine + Open Chain Glucose + H_2_OGlycine + Open Chain Glucose + H_2_O + Fe^2+^ + Fe^3+^ + H^+^ + OH^−^

Note that the dissociation of water into H^+^ and OH^−^ could equally be included as a reaction so that the initial reactant inputs could be simpler, but this would postpone reactions which are proton (H^+^) or hydroxyl (OH^−^) dependent by one iteration. Since this method is generally qualitative, this choice is simply expedient. Likewise, glucose can be input as its open chain or hemiacetal form, which interconvert in one reaction step. 

Reactions were expanded over 1–6 iterations, depending on the computational intensity each generated, which is a function of the ways the starting molecules’ functionalities interact with the employed reaction library as the network iterates. As noted above, reaction iteration typically causes exponential network growth, thus reaction networks were iterated so as to allow for the greatest diversity of products to be produced within the bounds of practical desktop computation. 

## 3. Results and Discussion

We first report the distribution of products from the modeled CRNs and how that distribution changes in the presence of iron species. We quantified the differences in terms of product counts, reaction counts, and the relative compositions of networks containing iron compared to networks lacking iron.

### 3.1. Product Diversity

The presence of Fe^2+^ or Fe^3+^ in the CRNRs increases the overall product count in all the modeled reactions (Table 1 and Figure 4), which is not surprising since the inclusion of novel mechanisms increases the number of possible reactions and products. The number of products formed in the iron-containing pyruvic acid CRNR after five generations contains ~2.6 more species than the iron-free network, making it the most iron-sensitive of the reaction networks studied here. The formose CRNR grew by ~38% after the inclusion of iron species. In the four generations of the glucose degradation reaction CRNR, iron increased product diversity by ~19%. In the three iterations of the Maillard reaction CRNR, the product count increased by ~14% with the inclusion of iron, making it the least iron-sensitive network studied.

Figure 5 shows curve fitting to the results which allows for extrapolation of the data to ranges beyond what is computationally tractable.

For all modeled reactions, product growth is roughly exponential, mainly due to the way new reaction mechanisms are able to be applied to new products. Curve fitting allows for the estimation of the product count for the glucose degradation and Maillard reaction networks beyond what can practically be computed. In general, all of these CRNRs grow roughly exponentially (by a factor of ~10) as a function of reaction generation. Iron-containing networks grow in diversity slightly faster than iron-free networks.

The mass constraint set on the networks, which limits the number of compounds that can be formed in each CRNR, is imposed for computational practicality. It should be borne in mind that such product mass constraints always eventually saturate since at some point all possible products within a constrained mass range are discovered, and if some product requires a higher molecular weight precursor, it will never form in these simulations. To examine this effect, we ran the formose reaction simulation for 15 generations with an upper mass limit of 100 as opposed to 200 AMU. Figure 6 confirms the saturation of the CRNR using this lower MW limitation.

Using a 100 AMU cutoff, the formose CRNR saturates after 13 generations because no new compounds can form without violating the product mass limit condition. In real-world chemistry, networks such as these can grow exponentially if there is no upper limit on the product mass as long as feedstocks are available. While we have not systematically performed the same analysis for ranges of AMU limits between 100 and 200 AMU, or beyond 200 AMU, we expect the output data to always find the limit of the species which can be discovered using these methods. Indeed, Figure 4A already suggests that saturation of reaction-discoverable chemical space does not occur after five generations for the formose reaction (contrast there being 52 novel product species after five generations using a 100 AMU cutoff vs. >30,000 for a 200 AMU cutoff). Interestingly, this perfunctory analysis suggests that mapping the generation and interconversion of low MW species in diversity generating reactions is already achievable, while that of even moderate MW species is still not computationally tractable, which may have implications for understanding the structure of modern metabolism. If the origin of life depends on interactions among molecules in the >200 AMU MW range (which would, for reference, already include such low complexity compounds as di- and tripeptides), it will not be easily predictable using these techniques and will require a more clever understanding of interfacial molecular recognition and reaction feedback mechanisms. 

### 3.2. Reaction Diversity

The compound nodes in a CRN are connected to each other by edges representing reaction mechanisms derived from real-world studies. The addition of iron-catalyzed mechanisms to the reaction rule database increases the number of rules that are applied in each reaction network. However, the magnitude of the increase varies depending on the input reactants.

In the formose reaction network, 32 encoded rules were applied over five generations, producing a total of 54,525 reaction edges in the absence of iron which increased to 81,049 edges after addition of the six encoded iron-containing rules (see Table 2). This is an ~49% increment in the number of reaction edges, while the compound count increased by only 38%. It should be made clear that some reactants enable the application of novel mechanisms more than others, and some compounds are inherently more reactive (e.g., more rules can be applied to them) than others. This increment is visible in the reaction count distribution across iterations, and indicates that some compound nodes are used multiple times (Figure 7).

In the pyruvic reaction network, the total reaction edge count increases from 17,446 to 45,367 by the inclusion of iron, an increase of ~160%, which is similar to the growth in the node count (~150%). Most of this increase occurs in the fifth generation.

In the four generations of the glucose degradation reaction network, the iron-containing network produces ~22% more reaction edges, growing from a total of 19,271 to 23,661 edges. 

In the three generations of the Maillard reaction, the edge count grows from a total of 8856 to 10,460 (an ~18% increase).

The addition of iron-containing reactions affects the edge distribution of the networks; however, this occurs unevenly due to the unique frequency with which different rule types can be applied to each unique reactant set in each generation. For example, the formose reaction network consists of 16% aldol condensation edges in an iron-free CRNR, but only 13% of the total edges are attributable to aldol condensations in the formose CRNR in the presence of iron. This decrease in edge count percentage is attributable to the contribution of new reaction rules added to the iron-containing network.

In the pyruvic acid reaction network, 31 unique reaction rules were applied 17,446 times in the absence of iron. In comparison, these same 31 rules applied together with six iron-specific rules produced 45,367 edges. Common rules in both the iron-free and iron-containing networks have comparable total edge counts, but the inclusion of iron lowers their relative percentage contribution to the overall network. For example, the aldol condensation rule makes up 17% of total rules applied in the iron-free network, while in the iron-containing network, this rule contributes only 16.5% of the total rule applications. 

The iron-containing glucose degradation network used a total of 38 rules producing 23,661 edges over three generations, which represents an ~20% increase in edge count over the iron-free network. In this network, two particular reaction rules (both variations of β-decarboxylation) were applied more than twice as frequently compared to the iron-free network. 

In addition to adding new edges and nodes to networks, the addition of iron also alters the relative frequency of rule application. In all the reaction networks, the reaction rules lacking iron increase because the application of iron-containing rules produces new substrates to which rules can be applied. That is, the iron rules “catalyze” other reaction mechanisms by providing substrates for their application. The addition of iron causes up to a 3.5× increase in the application frequency of non-iron-containing rules. Table 3 and Table 4 below summarize reaction rule applications across the networks.

### 3.3. Catalysis

To detect compounds most potentially affected by iron catalysis, the set of compounds produced in the iron-free network was compared with an iron-containing network to find the product overlap. A compound is defined as catalyzed if it is obtained in an earlier generation in an iron-containing network relative to an iron-free one since it requires fewer reaction steps to be produced. The results are shown in Table 5.

Among the networks studied, iron catalysis is most pronounced in the pyruvic acid reaction network, which likely depends on the nature of the input reactants and the interplay between the available reactive motifs and reaction rules. The percentage catalysis for the Glucose Degradation Reaction and Maillard Reaction is lesser compared to the other two networks because of their lesser numbers of generations. Furthermore, different reactants pose different reaction contexts leading to differences in catalysis intensity.

### 3.4. Graph Metric Comparisons

A CRN is directional, meaning reactant nodes point to product nodes. In a network graph, the degree of a node represents the number of edges it is connected to, which is, in some sense, a metric of the “importance” of that node [31]. Here, this could be interpreted as how often a given compound influences the chemistry of its neighbors. Since chemical reactions are directional in nature, the degree of nodes in reaction network graphs can also be measured in terms of their in-degree and out-degree. In-degree counts the number of edges pointing toward a node, and out-degree counts the number of edges pointing away from a node [31]. In-degree in a CRNR measures the number of times a node occurs as a reaction product, while out-degree measures the number of times a node is a reactant.

Comparing degree metrics of the iron-free and iron-containing networks probes the change in the relative importance of different compounds in the network. It also informs if a node is catalyzed by iron-dependent rules since in-degree quantifies the number of edges entering a node. Degree metrics for the top ten nodes for each network are shown in Figure 8.

As might be expected, water is generally the most connected node [32], and HCHO is usually a close second. Generally speaking, the input reagents benefit from “founder effects” when measuring graph degree metrics, i.e., simply by being the earliest members of the network, they accrue edges for more generations. This effect can also be seen in the iron-containing graphs in which Fe^2+^ and Fe^3+^ assume the third and fourth-place positions in terms of node degree in the Formose and Pyruvic Acid networks. Fe^2+^ and Fe^3+^ appear in the fifth and sixth positions in the Glucose Degradation network and in the sixteenth and nineteenth positions in the Maillard Reaction Network (see SI). Other important species as measured by node degree generally reflect how commonly applied reaction rules involve those species, and for the formose reaction network includes methanol, formic acid, and CO_2_, the first two of which are produced by the Cannizzaro reaction and the last by various decarboxylation reactions.

### 3.5. Gephi Visualization of the Networks

As the CRNRs are very large, containing tens of thousands of chemical species, Gephi- an open-source graph network visualization and exploration software package [27] offers a useful way to visualize their connectivity and development across generations. Gephi automatically optimizes the spatial layout of networks to enable their easy visualization, but as a result, iron-free and iron-containing CRNRs may not end up having similar layouts. Figure 9 shows a graphic representation of all nodes and edges of the CRNRs studied here. Nodes are colored according to the generation of the first appearance and sized by total node degree. For easy visualization, the edges of reactions involving iron species are colored red.

As can be seen, in iron-containing networks, the density and connectivity of networks grows considerably, as visible in the large number of nodes and their general tighter packing. The red-colored regions in Figure 9 highlight iron-involving reactions. It should be noted that these plots do not reflect reaction yields or kinetics, though node degree may reflect these properties.

It is important to note that for the reasons described above, the number of potential products generally grows exponentially in CRNRs, at least where they are not capped in their exploration by imposing growth limits. This being the case, the initial input mass of reagents would need to be spread over more and more compounds, suggesting these types of complexity-generating reactions should end up with a very large set of very rare products. Generally speaking, this is the case (see, for example, [18,20]), though there is also usually a gradient of products in terms of abundance, such that a handful of product compounds may account for a major fraction of the mass balance (see for example [29,33]). The ability to detect the major or minor components of product mixtures typically hinges on the analytical tools employed, e.g., abundant compounds may be most easily tracked using various types of chromatography and NMR spectroscopy, while the abundance of rare species may be better tracked by “one-shot” techniques like electrospray ionization mass spectrometry (e.g., [18]).

It is apparent from these plots and the various other analyses presented above (see, for example, Table 5) that iron ions can have significant though variable effects on the course of complex reactions. Even so, they may not completely govern the overall evolution of such networks, depending on the placement and frequency of individual iron-steered reactions.

## 4. Conclusions

This work presents an effort to explore in silico how the addition of metal catalysis (specifically iron) may modify the evolution of CRNRs. Here, reaction mechanisms involving iron added were vetted using literature review as enabled by the Scifinder database. While this database is extensive, it is likely that data mining under-represents the extent of iron-catalyzed small molecule catalysis, as there likely remain many undiscovered and unreported iron-involving reaction mechanisms. Furthermore, this type of modeling does not take into account the effects of pH, temperature, or concentration on the relative rates of reactions, all of which can significantly skew relative product concentrations, it merely presents a road map of possible reactions. As such, such models offer good first-order ways to explore potential CRN product diversity, which can be useful for experimental studies. 

Nevertheless, such CRNRs may discover the most obvious and likely robust reaction mechanism network effects since many of the underlying mechanisms have already been discovered. These CRNRs can also be queried to find autocatalytic reaction loops (see [20]) and reaction sequences which cycle iron between its +2 and +3 redox states to effect autocatalytic outcomes. We plan to explore the aspects of these CRNRs in future work. As such, the in silico exploration of complex abiotic organic reaction sequences which involve iron species offers a way to explore how early chemical evolution became coupled to inorganic geochemistry.

## Figures and Tables

**Figure 1 molecules-27-08870-f001:**
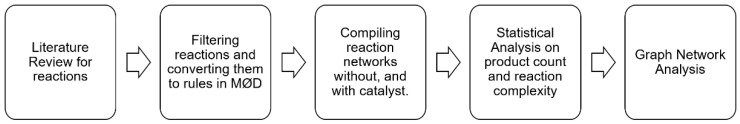
Overview of the workflow explored here. Generic organic transformations mediated by iron were collected from the experimental chemical literature and converted to machine readable form. These were then used to construct CRNR using the MØD software package. These modeled CRNRs were then analyzed with regard to their products and network topology using computational tools.

**Figure 2 molecules-27-08870-f002:**
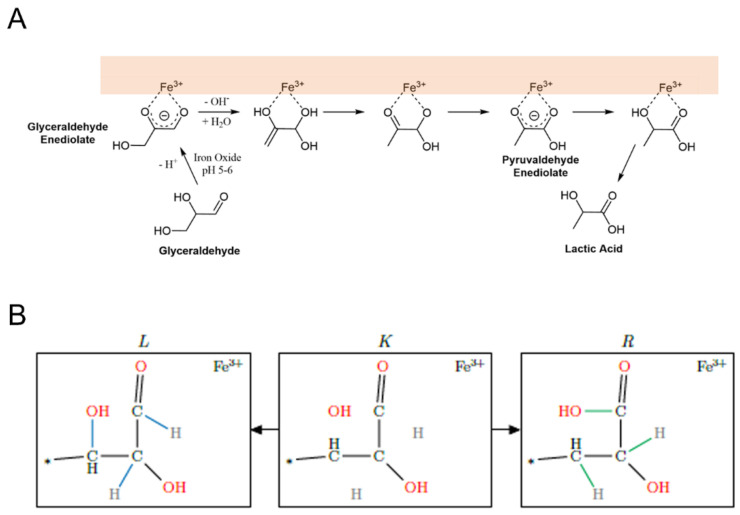
(**A**). proposed mechanism for the observed conversion of glyceraldehyde to lactic acid in the presence of mineral surface-bound iron (III) (mineral surface indicated by the orange box) adapted from [11]. In this reaction iron plays the role of catalyst, as it is not consumed or altered during the reaction; (**B**). A generic representation of the reaction mechanism presented in (**A**) in the MØD-based workflow. The MØD algorithm looks for compounds with substructures matching L and transforms the bonds between them as per the rule to produce the substructure in R. The context K is analogous but not identical to the “intermediate” of a conventional reaction mechanism. Here, bonds colored in blue represent bonds broken in the course or the reaction and those in green are bonds created by the reaction. Atom coloring is merely a guide for the eye. Asterisks represent “wild card” atoms which may be of any specified type, allowing such mechanisms to be generally applied to any molecule containing the specified substructure. In this reaction mechanism Fe^3+^ is a catalytic species that must be present for the Fe^3+^-catalyzed reaction to proceed, but which is not altered or consumed in the reaction. An equivalent “uncatalyzed” reaction mechanism can be written lacking Fe^3+^.

**Figure 3 molecules-27-08870-f003:**
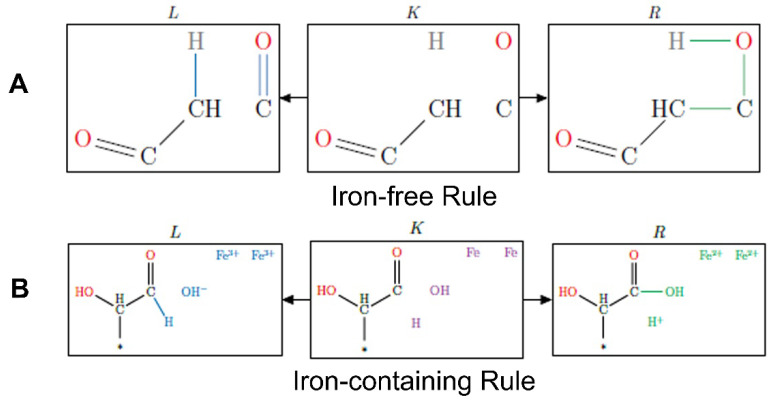
Examples of iron-independent (**A**) and iron-dependent (**B**) reactions as encoded in MØD. (**A**). Aldol condensation reaction; (**B**). A two-electron oxidation of an aldehyde accompanied by the reduction of two iron (III) atoms to two iron (II) atoms.

**Figure 4 molecules-27-08870-f004:**
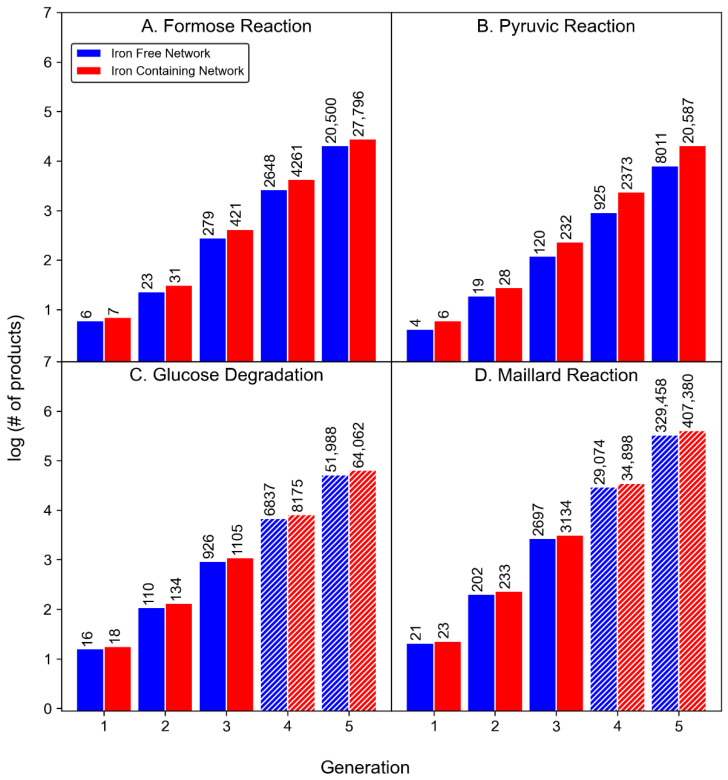
Plots showing the measured and projected increase of products in four model reactions in the absence (blue) or presence (red) of iron species as a function of reaction generation. The hatched bars in plots **C** and **D** show the predicted number of compounds for generations beyond those computed.

**Figure 5 molecules-27-08870-f005:**
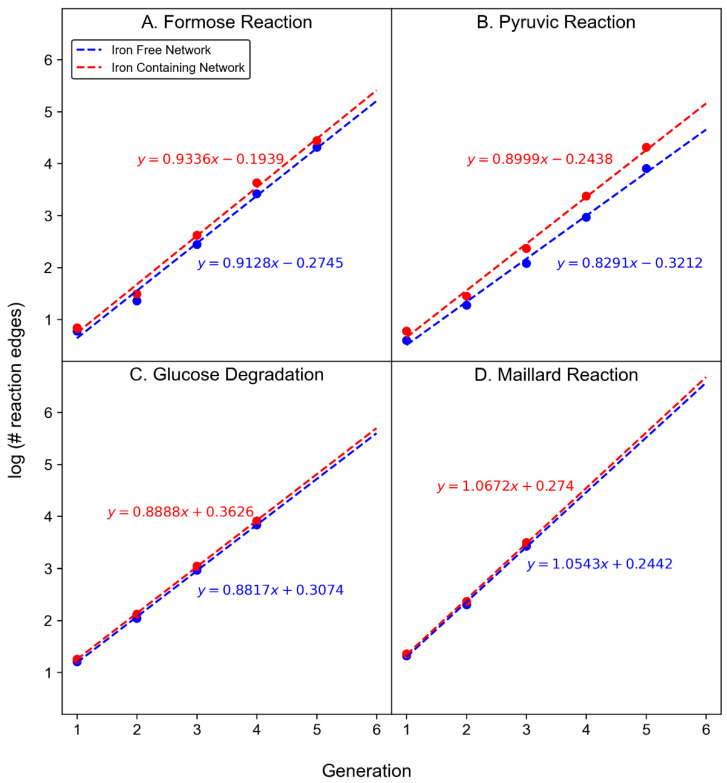
Log number of products vs. reaction generation for each reaction for iron-containing (red) or iron-free (blue) CRNRs. Data points are computed values, dashed lines are best fits to the computed data points.

**Figure 6 molecules-27-08870-f006:**
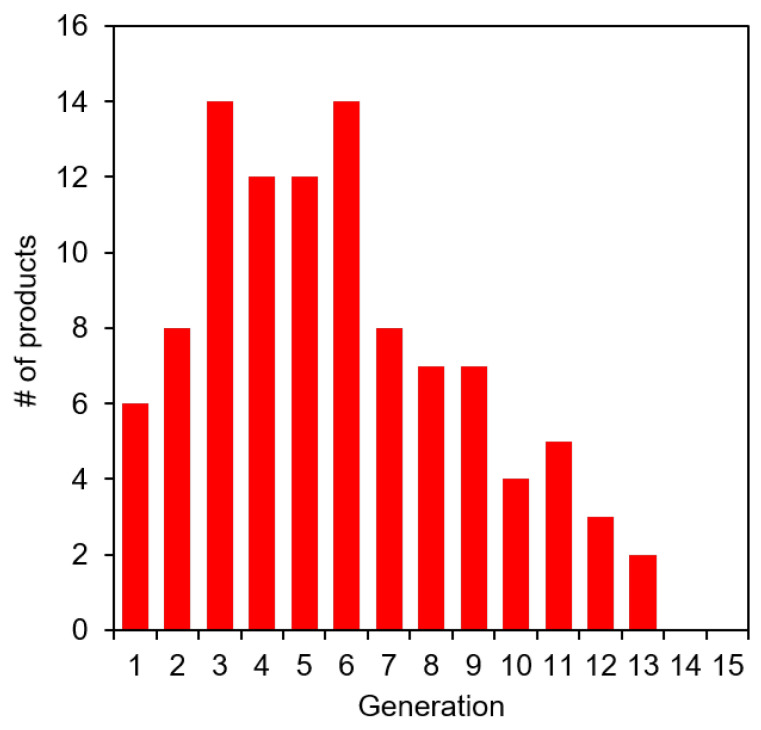
Modeled product counts for 15 generations of the iron-catalyzed formose reaction CRNR with an imposed upper mass limit of 100 AMU. After generation 13, no new compounds are discoverable in this network.

**Figure 7 molecules-27-08870-f007:**
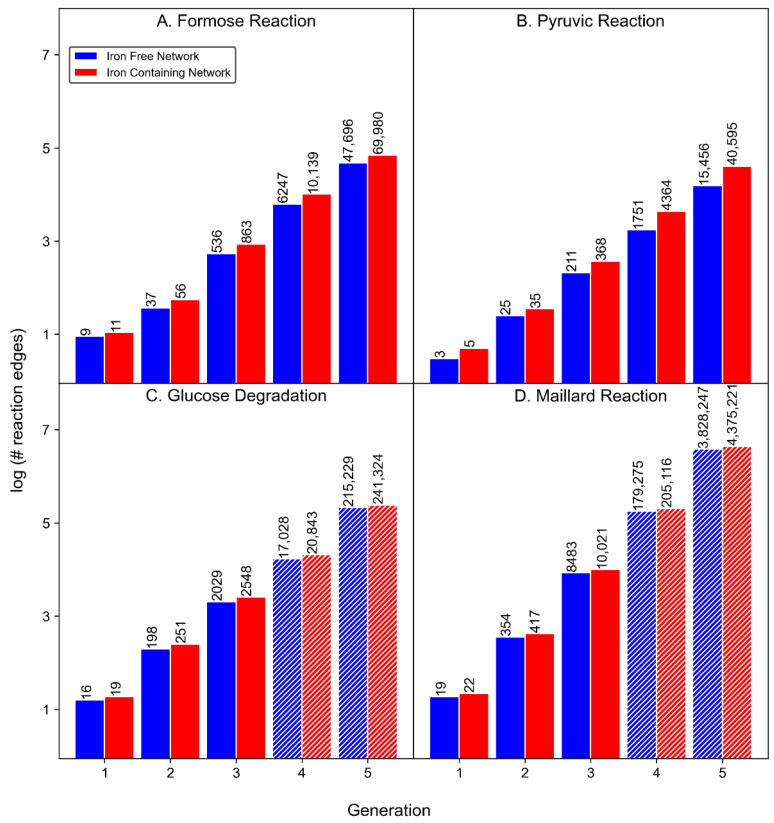
Reaction edges vs. reaction generation in the four model reactions in the absence (blue) or presence (red) of iron species. In plots **C** and **D**, hatched bars represent the predicted number of reactions in generations four and five.

**Figure 8 molecules-27-08870-f008:**
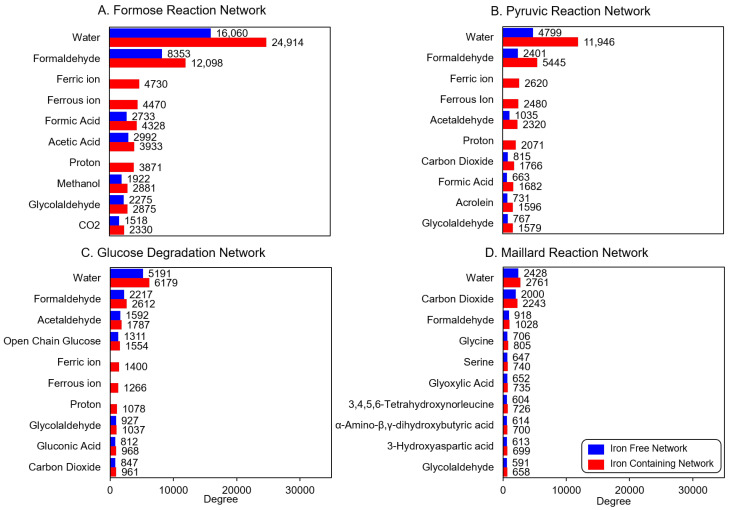
Total (in- plus out-) degree comparisons the studied reaction networks. Blue bars: iron-free reactions, red bars: iron-containing reactions. With few exceptions, the order of the degree of comparison does not change significantly in the networks. The magnitude of the degree for the common compounds of the networks always increases in the iron-containing network. Note that the comparisons are drawn only for the ten highest-degree compounds of the iron-containing network. Refer to Appendix A for the degree of all products produced in all the networks.

**Figure 9 molecules-27-08870-f009:**
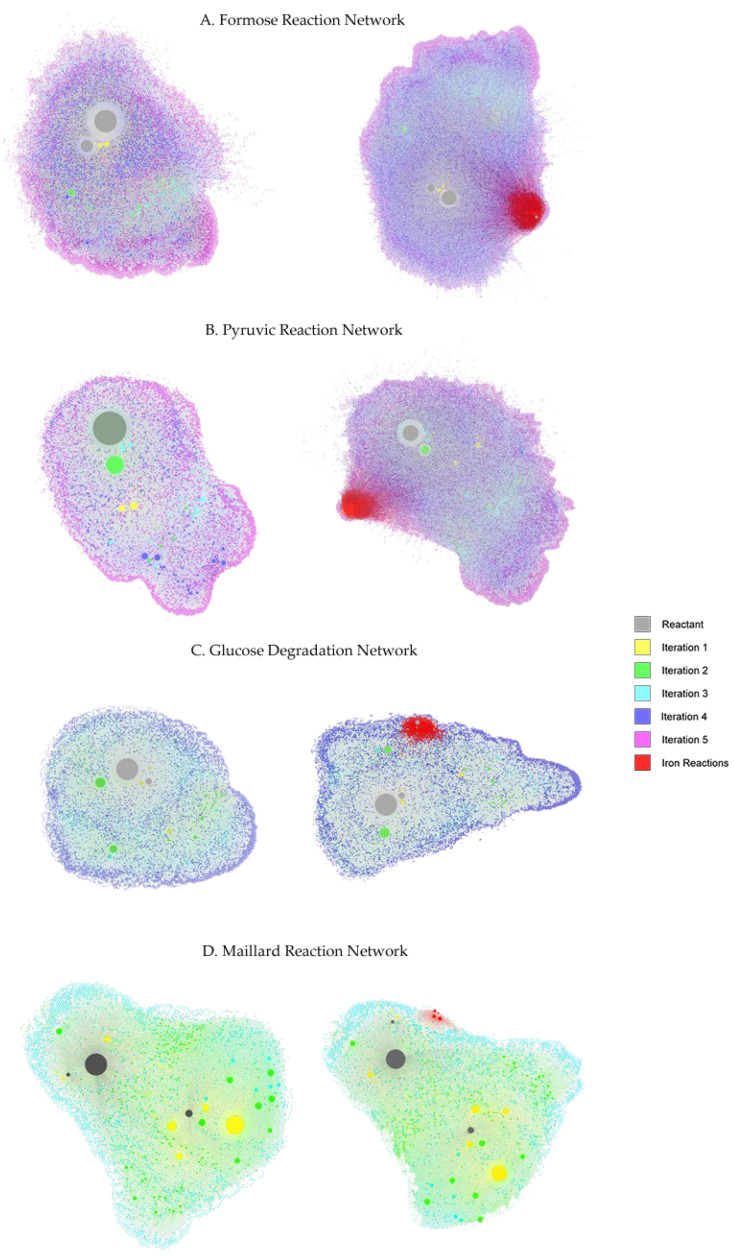
Gephi visualization of modeled reaction networks. The left plots are iron-free networks and the right plots are the iron-containing networks for each reaction. Iron-utilizing reaction edges are colored in red.

**Table 1 molecules-27-08870-t001:** Total number of products produced in the networks studied.

Reaction	# of Iterations	Iron-Free Network	Iron-Containing Network
Formose Reaction	5	23,459	32,523
Pyruvic Acid Reaction	5	9081	23,232
Glucose Degradation Reaction	4	7891	9438
Glucose-Glycine Maillard Reaction	3	2923	3417

**Table 2 molecules-27-08870-t002:** Total frequencies of reaction-rule application in the studied networks.

Reaction	# of Iterations	Iron-Free Network	Iron-Containing Network
Formose Reaction	5	54,525	81,049
Pyruvic Acid Reaction	5	17,446	45,367
Glucose Degradation Reaction	4	19,271	23,661
Glucose-Glycine Maillard Reaction	3	8856	10,460

**Table 3 molecules-27-08870-t003:** Frequencies of reaction rule applications in the formose reaction network. (% change = (frequency in the iron-containing network/frequency in the iron-free network) × 100). The addition of iron to the formose reaction network adds up to 2365 new unique iron-induced reactions, but also provides new products that can react with other compounds in the network using iron-independent reaction mechanisms, changing the frequency of all non-iron reaction mechanisms.

Reaction Name	Iron-Free Network	Iron-Containing Network	% Change
β-γ Unsaturated Acid Decarboxylation	12	54	350%
2. α-β Unsaturated Acid Decarboxylation	5	17	240%
3. α-Keto Acid Decarboxylation	78	187	140%
4. Ring Closure 5 membered O, O	398	946	138%
5. Michael Addition 0.2, Inverse	221	520	135%
6. Ring Closure 6 membered O, O	227	527	132%
7. Ring Closure 7 membered O, O	82	190	132%
8. β Decarboxylation	106	222	109%
9. Benzilic Acid Rearrangement, Inverse	365	670	84%
10. Benzilic Acid Rearrangement	326	592	82%
11. Knoevenagel H, Inverse	1830	3189	74%
12. Canizzaro	242	417	72%
13. Hemiacetal Formation for 5 membered rings, Inverse	348	586	68%
14. Hemiacetal Formation for 5 membered rings	2482	4082	64%
15. Canizzaro 2, HCHO (oxidation)	2731	4326	58%
16. Hemiacetal Formation for 6 membered rings, Inverse	135	211	56%
17. Keto-enol migration twice	1884	2936	56%
18. Retro Aldol	3756	5849	56%
19. Elimination + enol to keto	3825	5926	55%
20. Hemiacetal Formation for 6 membered rings	1500	2320	55%
21. Canizzaro 2, HCHO (reduction)	1691	2545	51%
22. Hemiacetal Formation for 7 membered rings	599	896	50%
23. Knoevenagel C, Inverse	1623	2384	47%
24. Elimination2	3966	5758	45%
25. Hydration of C=C(O)	522	753	44%
26. Hemiacetal Formation for 7 membered rings, Inverse	52	75	44%
27. Knoevenagel H	5494	7576	38%
28. Hydration of C(=O)C	1731	2359	36%
29. Michael Addition 0.2,	4327	5508	27%
30. Knoevenagel C	5228	6484	24%
31. Aldol Condensation	8739	10,579	21%
32. Iron-induced Rules	0	2365	-
a. Fe^+2^ to Fe^+3^, α Keto Reduction of a Carboxylic Acid	0	187	-
b. Fe^+2^ to Fe^+3^, Conversion of α Hydroxy Group of an aldehyde to α Keto group, Inverse	0	636	-
c. Fe^+2^ to Fe^+3^, Conversion of α Keto Group of an Acid to α Hydro group	0	183	-
d. Fe^+3^ to Fe^+2^, Aldehyde to acid	0	412	-
e. Fe^+3^ to Fe^+2^, Conversion of α Hydro Group of an Acid to α Keto group	0	251	-
f. Fe^+3^ to Fe^+2^, Conversion of α Hydroxy Group of an aldehyde to α Keto group	0	566	-
g. Fe^+3^, β elimination from a sugar & Conversion to Acid	0	130	-

**Table 4 molecules-27-08870-t004:** Frequencies of reaction rule applications in the other three studied networks. The top five reactions most enhanced by the presence of iron are shown. An expanded version of this table is attached in the Appendix A.

Network	Reaction	Iron-Free Network	Iron-Containing Network	% Change
**Pyruvic** **Reaction**	Ring Closure 7 membered O, O	40	194	385%
	Hemiacetal Formation for 7 membered rings	87	321	269%
	Ring Closure 6 membered O, O	170	543	219%
	Hemiacetal Formation for 6 membered rings	297	925	211%
	Michael Addition, Inverse	318	979	208%
**Glucose** **Degradation** **Reaction**	β-γ Unsaturated Acid Decarboxylation	2	6	200%
	α-β Unsaturated Acid Decarboxylation	3	8	167%
	α-Keto Acid Decarboxylation	11	19	73%
	Ring Closure 7 membered O, O	29	49	69%
	Benzilic Acid Rearrangement, Inverse	65	103	58%
**Glucose-Glycine Maillard Reaction**	Strecker Degradation Dicarbonyl, C, H, C, H	52	88	69%
	Strecker Degradation Dicarbonyl, C, H, H, C	52	88	69%
	Strecker Degradation Dicarbonyl, H, H, C, H	13	22	69%
	Strecker Degradation Dicarbonyl, H, H, H, C	13	22	69%
	Amide Formation Hydrolysis, C	9	15	67%

**Table 5 molecules-27-08870-t005:** Comparison of the number of compounds catalyzed by iron species in each CRNR. Percentage Catalysis = Number of compounds catalyzed/ Total Number of compounds in the iron-free reaction network.

CRN	# Generations	# Catalyzed Compounds	% Catalyzed
**Formose Reaction**	5	1634	7%
**Pyruvic Reaction**	5	765	8.4%
**Glucose Degradation Reaction**	4	231	3%
**Glucose-Glycine Maillard Reaction**	3	33	1.1%

## Data Availability

Not applicable.

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
