# Peer review of "The Effects of Iron on In Silico Simulated Abiotic Reaction Networks"

_molecules, 2022, doi:10.3390/molecules27248870_

Round 1

Reviewer 1 Report

This paper uses computational methods to study chemical reaction networks that might occur prebiotically and to determine whether Fe2+ and Fe3+ ions are likely to play an important role in catalyzing the reaction networks. This seems like an interesting and well-motivated question. However, from reading the paper, I am unsure about the methods and I am unsure whether the conclusions are justified.

The paper uses software called MOD that was developed by other authors and which is not well explained in this paper. What are the rules used by the software to determine if a reaction can occur? Presumably it must satisfy the rules of chemistry in some sense? But can we really specify these well enough to write computational rules? In any case, this needs to be explained in the paper.

The rules used appear to be quite permissive and inclusive, so that large networks of reactions are built up. I presume this allows many reactions that are at least theoretically possible, but whose reaction rate may very well be vanishingly small. As far as I can see, there is no attempt by the software to assign a reaction rate. I would presume that the relevance of iron in prebiotic chemistry is that it can catalyze many reactions – so these reactions will occur much faster in the presence of iron than in absence. But I don’t think this question can be answered with the current analysis. Is there a way to exclude reactions that we expect to be unimportant because they are too slow? If we could somehow count just the fast reactions, then maybe the effect of iron on the number of fast reactions would be much larger?

The example reactions in Fig 3 are not very clear. It looks like we have incomplete molecules – Carbon bonds don’t add up to 4 etc. How do we understand these reaction pictures?

Also Fig 3 is supposed to show an iron-dependent and an iron-independent reaction. But these reactions are different. I was expecting to see an example where iron could catalyze a reaction that could also occur without iron. I am confused! What does it mean to say that iron is a catalyst? In Fig 3, Fe3+ is reduced to Fe2+. So how is a cycle maintained? There must be a reaction that gets back to Fe3+, otherwise Fe3+ is consumed and it is not a catalyst. How does this reaction network deal with maintaining the overall oxidation state of iron? Also, I thought that the origin of life was in anaerobic conditions and that iron would be mostly Fe2+. The Fe2+ would then be a reducing agent for organic molecules. So most reactions would involve changing Fe2+ to Fe3+, not the reverse, as is illustrated here. How does this work fit into the picture of an anaerobic world at the time of origin of life?

There is a mass limit on the size of molecules that can be synthesized. This seems essential in order to avoid combinatorial explosion of the number of reactions. However, it seems to me that the explosion is really in the number of theoretically possible reactions rather than the number of important reactions that would really happen at an appreciable rate. Maybe very complex multi-atom molecules simply would not arise in real chemistry? However, imposing a mass limit seems to exclude the possibility of forming polymers. Polymers such as proteins, nucleic acids and polysaccharides are essential in biochemistry. But the formation of polymers involves repeating the same reaction many times (peptide bond formation or phosphodiester bond formation). Adding each monomer is not a separate reaction. Is there a way to modify the software to allow formation of polymers from small sets of monomers? This would allow large molecules without an explosion of the number of reactions. I would guess that polymers of small building blocks are more relevant than very complex monomers with large numbers of atoms.

In summary – this paper seems a promising start with a new approach, but there are many things that are unclear, and I think it requires a lot of work in order to justify what was done and make the conclusions convincing.

Author Response

Reviewer 1 Comment Responses

This paper uses computational methods to study chemical reaction networks that might occur prebiotically and to determine whether Fe2+ and Fe3+ ions are likely to play an important role in catalyzing the reaction networks. This seems like an interesting and well-motivated question. However, from reading the paper, I am unsure about the methods and I am unsure whether the conclusions are justified.

We appreciate the reviewer’s comments. Combined with some of the comments from Reviewer 2, we think it was not made explicitly clear that the underlying logic and methods of this approach have been published previously (Sharma et al. 2021, Arya et al., 2022), thus we did not elaborate them extensively. We have made efforts to clarify this for the reader as detailed below.

The paper uses software called MOD that was developed by other authors and which is not well explained in this paper. What are the rules used by the software to determine if a reaction can occur? Presumably it must satisfy the rules of chemistry in some sense? But can we really specify these well enough to write computational rules? In any case, this needs to be explained in the paper.

We have added clarifying text in the introduction, including several paragraphs which explain the workflow in more detail, as well as a weblink to the software on github.

These CRNRs are computed using reaction rules that are in-silico representations of real-world chemical reaction mechanisms. The rules used appear to be quite permissive and inclusive, so that large networks of reactions are built up. I presume this allows many reactions that are at least theoretically possible, but whose reaction rate may very well be vanishingly small. As far as I can see, there is no attempt by the software to assign a reaction rate. I would presume that the relevance of iron in prebiotic chemistry is that it can catalyze many reactions – so these reactions will occur much faster in the presence of iron than in absence. But I don’t think this question can be answered with the current analysis. Is there a way to exclude reactions that we expect to be unimportant because they are too slow? If we could somehow count just the fast reactions, then maybe the effect of iron on the number of fast reactions would be much larger?

Indeed, as detailed in Arya et al. 2022, these approaches are not able to assign or estimate absolute reaction rates, rather they depend on literature precedent for their occurrences under mild aqueous conditions for their inclusion. The reaction rules are selected based on literature precedent for their proceeding at noticeable speed in water at room temperature. In Arya et al. (2022) the free energy changes of these rule transformations were evaluated. The large majority fall within the range of +50 to -50 kJ mol-1 (~ +12 to -12 kcal mol-1) which would give Keq values of ~ 1.2 to 8.0 in the forward direction. A small fraction are outside of this range, corresponding to reactions with more extreme Keq values. In any event such analysis is possible, and will be the subject of future work. 

The example reactions in Fig 3 are not very clear. It looks like we have incomplete molecules – Carbon bonds don’t add up to 4 etc. How do we understand these reaction pictures?

This has been corrected. Kindly refer to the caption of Figure 2.

Also Fig 3 is supposed to show an iron-dependent and an iron-independent reaction. But these reactions are different. I was expecting to see an example where iron could catalyze a reaction that could also occur without iron. I am confused! What does it mean to say that iron is a catalyst? In Fig 3, Fe3+ is reduced to Fe2+. So how is a cycle maintained? There must be a reaction that gets back to Fe3+, otherwise Fe3+ is consumed and it is not a catalyst. How does this reaction network deal with maintaining the overall oxidation state of iron? Also, I thought that the origin of life was in anaerobic conditions and that iron would be mostly Fe2+. The Fe2+ would then be a reducing agent for organic molecules. So most reactions would involve changing Fe2+ to Fe3+, not the reverse, as is illustrated here. How does this work fit into the picture of an anaerobic world at the time of origin of life?

The reaction rules shown are only a few of the many rules available for these reaction networks. There are reaction rules in which Fe+2 is converted back to Fe+3. We have added brief discussions of the questions of iron redox cycling and catalysis.

There is a mass limit on the size of molecules that can be synthesized. This seems essential in order to avoid combinatorial explosion of the number of reactions. However, it seems to me that the explosion is really in the number of theoretically possible reactions rather than the number of important reactions that would really happen at an appreciable rate. Maybe very complex multi-atom molecules simply would not arise in real chemistry? However, imposing a mass limit seems to exclude the possibility of forming polymers. Polymers such as proteins, nucleic acids and polysaccharides are essential in biochemistry. But the formation of polymers involves repeating the same reaction many times (peptide bond formation of phosphodiester bond formation). Adding each monomer is not a separate reaction. Is there a way to modify the software to allow formation of polymers from small sets of monomers? This would allow large molecules without an explosion of the number of reactions. I would guess that polymers of small building blocks are more relevant than very complex monomers with large numbers of atoms.

We have added discussion of these points where the concept of mass limitation is discussed in the main text now, and discussed the issue of polymer computation.

In summary – this paper seems a promising start with a new approach, but there are many things that are unclear, and I think it requires a lot of work in order to justify what was done and make the conclusions convincing.

We thank the reviewer for their comments, we hope the revised version adequately addresses their concerns. We appreciate that some aspects of this approach can be hard to appreciate immediately, but we hope we have managed to make them more clear for the reader.

Reviewer 2 Report

This manuscript by Shahi and Cleaves describes the effect of including iron in various prebiotically simulated chemical networks. The motivation for this work is that iron is likely present on the early Earth and may have played a role in prebiotic chemistry. This study examines the role of iron in expanding/diversifying chemical networks. 

This topic is certainly of interest, and is quite novel and challenging to address computationally. There are certainly some interesting results provided, but the presentation and clarity of the manuscript could be significantly improved. 

Major comments:

-The main results/conclusions of the paper are somewhat difficult to parse in the current form of the manuscript. Perhaps this is because there are so many numbers and ways to quantify these various reaction networks. Overall, simplifying the text to explicitly state core results in a straightforward manner, changing some of the paragraphs that are numbers-heavy into tables, and offering the reader a bit more of a narrative throughout the analysis would improve the manuscript significantly. Do the results/conclusions support the last statement in the abstract? This statement is difficult to parse given only the information in the abstract. For example, what does diversifying on the order of 20% mean? What are kinetically-naive models? These terms and numbers don’t seem to be repeated in the manuscript conclusions or main results. 

-MOD is referred to in the text quite a few times before it seems to be properly introduced. For example, it appears in Figure 1, Figure 2, Figure 3 line 101. The first definition of MOD (if I’m correct) is in the paragraph beginning on line 110. Please define MOD when it first appears. Perhaps as a more general comment: a lot of the complexities of reaction networks and analysis were difficult to comprehend for the non-expert. Any efforts to remove jargon would be beneficial. 

-Figure 2 - Panel B needs to be described more, both in the caption and the text. For example, what do L, K, and R refer to? What is the significance of the colored bonds? Can you explain the direction of the arrows? Also, MOD still hasn’t been defined. Perhaps adding a section to the text describing these types of diagrams would be useful. 

-The terminology changes from “iron-exclusive/iron-inclusive” to “Fe(+)/Fe(-)” throughout the manuscript. Please choose a consistent terminology. “Iron-exclusive” is confusing to me. One possible alternative might be “iron-containing” and “iron-absent”? Iron (-) and iron (+) are introduced around line 270. Iron-free and iron-inclusive are used in Fig 9 caption. Regardless of what is chosen, it should be consistent throughout. 

-Figure 3 - Consider labeling the top row and bottom row as “No iron” and “With iron” (or whatever terminology you decide upon). Also, this figure needs an explanation for L, K, R and the different bond/atom colors. 

-Why does an upper mass limit of 200 units accurately capture the system? The authors present a test case with a 100 unit upper limit, but could provide a deeper justification for their choice of 200 units as an upper limit. If it’s purely a computational limit, please state this and justify why the results presented are still justified to understand these systems. 

-A number of section in the results/discussion where many numbers are listed for the various reactions might be better presented as tables. For example, can the information in lines 163-169 be converted into a table showing reaction, # of iterations, # of products without iron, # of products with iron? 

-Figure 4 - It would really help the reader to include a legend indicating what the blue and red colors represent (rather than just having this information in the caption). The font size everywhere (tick labels, axis labels, titles) is too small (and panel A title looks to have inconsistent font size). Also, the explanation for the calculation of the predicted number of compounds for panels C and D comes after the figure is referenced in the text, which is not ideal (I think the explanation appears in line 180). 

-Figure 5 - A legend would be helpful here as well. Also, please show the actual data as points without connecting lines and the fits with dotted/dashed lines. Consider color coding the equations of fit as red or blue. Axis labels, titles, and tick labels should be larger in size. 

-Figure 6 - Is the y axis showing the number of new products with each generation or the number of unique products? The labeling is different between Figs 4/5 and 6. Is this on purpose, or are these figures showing different quantities? Please make this explicitly clear in the text and captions if so. 

-Figure 7 - Include a legend and increase font size

-Table 1 - need to label what number is being shown in the columns under “Iron -“ and “Iron +.” Also, please use consistent terminology to refer to these cases throughout the manuscript. 

-Table 2 - Why do you choose to show all the reactions for the Formose reaction in Table 1, but only show the top five reactions for the other networks studied in Table 2? Please explain why this comparison is justified. Should equivalent tables perhaps be shown for the other three reaction networks in the SI? Also, the number being shown under “Iron -“ and “Iron +” needs to be labeled. I’m also wondering why the numbers in Table 2 are so different in magnitude than those shown in Table 1 (e.g. ~10000 vs. ~10)? 

-Figure 8 - add a legend for red and blue colors; use consistent terminology in caption; increase font size everywhere. Maybe briefly describe what “degree” means in words in the caption? 

Minor comments: 

-line 9 - Corresponding author and email need to be updated

-lines 17-18: “reaction of pyruvic acid” Into what? This phrasing isn’t parallel to the other three reactions mentioned. 

-line 37 - does STP need to be defined? 

-line 37 - possibly switch “energy change” to “energy difference”?

-line 48 - Fe^3+ - superscript is missing

-line 54 - provide a brief description of the Maillard reaction here?

-lines 60-62 - The wording of the sentence beginning with “Chemical transformations…” is complicated and difficult to parse. Perhaps rephrase as:  "Chemical transformations that are possible in the absence of iron may also be catalyzed by iron species; similarly, reactions that are impossible in the absence of iron may become possible when iron is present." 

-line 63 - “highly diversity-generating reactions” - complicated wording - consider rephrasing, perhaps as: “…includes a number of reactions that generate high chemical diversity, seeded by simple likely prebiotic reactants, including..."

-line 65 - Suggest breaking the sentence up:  "...among others. These reactions may produce hundreds to millions..."

-lines 70-71 - Can the authors provide another sentence of two elaborating on the graph-theory techniques and CRNRs here? This background seemed to be a little lacking throughout the introduction for the non-expert reader. 

-line 74 - semicolon doesn’t seem to be grammatically correct here. Rephrase? 

-line 87 - can the authors provide a rationale for choosing 25 degrees C as the lower bound on temperature? Cooler environments are occasionally suggested as plausible for the early Earth. 

-line 88 - Define Fenton reagents here?

-line 91 - delete “of course”?

-line 93-96 - Why are these two reactions specifically mentioned here? Are they outliers or difficult to classify for some reason? If so, please state this explicitly. 

-line 126 - reaction of concentrated pyruvic acid with what? Itself? Other chemicals? The description here isn’t in parallel structure to the other reactions described. 

-line 150 - “Reactions were expanded over 1-6 iterations…” It isn’t immediately clear what this means. Can you provide a general description of this process? 

-line 155 - This seems like an odd first sentence to the results and discussion section. Perhaps start with something more general and descriptive of the overall section to help guide the reader? 

-line 182 - need to specify what grows by a factor of ten - number of products, right? “e.g. the number of products in each generation grows by a factor of ten.” 

-line 184-185 - this sentence seems to be missing a word in order to make sense grammatically? 

-line 201-202 - Add a sentence comparing these results to the 200 AMU mass limit results for the five generations you followed there. It seems like the mass limit plays a huge role in the number of (new?) products - it’s ~10 for 100 AMU limit, but ~20000 for 200 AMU limit? Is this expected? 

-line 227-228 - Where are these numbers coming from? They don’t seem to precisely match the 5th generation numbers in panel B of Figure 7. 

-lines 230-232 - Again, these numbers don’t seem to match what is shown in Figure 7. Also, consider turning this paragraph into a table, or alternatively just qualitatively explaining the trends (e.g. with iron, the number of reactions edges is XX-YY% more than when iron is not included). 

-lines 240-247 - I'm confused by the overall point of this paragraph, perhaps due to the many numbers presented. Is the point that the iron-free and iron-containing networks are dominated by the non-iron containing rules? If so, what is the meaning of the increased diversity in the networks with iron present? 

-lines 250-252 - These two sentences seem to contradict each other. Too many numbers. Distill down to the main point please. 

-line 259 - change “increment” to “increase”?

-line 271-273 - This sentence has confusing wording. Consider revising. 

-line 277 - It seems like the pyruvic acid and Formose reactions are quite different in % catalyzed than the other two reactions. Can you offer speculation as to why this might be? 

-lines 302-304 - That’s only for pyruvic acid and Formose reaction, though, right? They also appear fairly high in glucose degradation, but not at all in the Maillard reaction. Why might this be? 

-line 312 - define Gephi

-line 316 - remove “below” 

-line 319-320 - This wasn’t immediately obvious to me when first looking at the plots. Can you offer the reader a bit more guidance as to what they are looking for/at? 

-line 324 - change “CNRNs” to “CRNRs”

Lines 333-336 - Is this the key point of the study? If so, I think it could be made a bit more explicitly throughout the results/discussion section. Perhaps because of the many numbers listed, it’s hard for the reader to parse the information and follow the main conclusions? Please also make sure this statement is consistent with the last sentence of the abstract. 

-line 338 - The conclusion section uses the words “in silico” a fair amount, when this has only been sparsely mentioned before. Can you make this more consistent throughout the paper? 

-line 338-339: “… metal, specifically iron, catalysis…” wording is a little odd. Perhaps revise to: “…how the addition of metal catalysis (specifically iron) can modify the evolution of CRNRs.” 

-line 341-343 - So what guarantees do you have that your conclusions here are representative of the general case with additional reactions/rules? 

-line 344 - how might pH, temperature, and concentration potentially affect these results? What uncertainties do they imply for your conclusions here? 

-line 346 - The sentence starting with “Nevertheless” is very long. Consider breaking it into multiple sentences. 

-line 348-349 - Regarding the statement: “…these CRNRs can be queried to find autocatalytic reaction loops and reaction sequences which cycle iron between its +2 and +3 redox state…” - this is the first mention of this as far as I can tell. Was this done at all in your study? Or could it be invoked in the future? How would this be done?

-line 352 - reference to organic geochemistry here felt like it was just thrown in at the end, without much actual motivation for it throughout the paper.

Author Response

Reviewer 2 Comment Responses

This manuscript by Shahi and Cleaves describes the effect of including iron in various prebiotically simulated chemical networks. The motivation for this work is that iron is likely present on the early Earth and may have played a role in prebiotic chemistry. This study examines the role of iron in expanding/diversifying chemical networks. 

This topic is certainly of interest, and is quite novel and challenging to address computationally. There are certainly some interesting results provided, but the presentation and clarity of the manuscript could be significantly improved. 

 We appreciate the time the reviewer has taken to prepare such a detailed review. We provide detailed responses below.

Major comments:

-The main results/conclusions of the paper are somewhat difficult to parse in the current form of the manuscript. Perhaps this is because there are so many numbers and ways to quantify these various reaction networks. Overall, simplifying the text to explicitly state core results in a straightforward manner, changing some of the paragraphs that are numbers-heavy into tables, and offering the reader a bit more of a narrative throughout the analysis would improve the manuscript significantly. Do the results/conclusions support the last statement in the abstract? This statement is difficult to parse given only the information in the abstract. For example, what does diversifying on the order of 20% mean? What are kinetically-naive models? These terms and numbers don’t seem to be repeated in the manuscript conclusions or main results. 

We thank the reviewer for this general observation. We have attempted to clarify the text and make the approach, results and conclusions more easily comprehensible to the reader. We have attempted to clarify what is meant by diversification both in terms of reactions and products, and pushed some of the numerical results to table format.

-MOD is referred to in the text quite a few times before it seems to be properly introduced. For example, it appears in Figure 1, Figure 2, Figure 3 line 101. The first definition of MOD (if I’m correct) is in the paragraph beginning on line 110. Please define MOD when it first appears. Perhaps as a more general comment: a lot of the complexities of reaction networks and analysis were difficult to comprehend for the non-expert. Any efforts to remove jargon would be beneficial. 

We appreciate that this is a similar concern raised by Reviewer 1, and that it was implicit that much of the methodological detail and justification was provided in a previously published paper that was cited in the text (Arya et al. 2022). To make this manuscript more easily understandable with referencing this prior publication, we have elaborated the methods and workflow considerably here.

-Figure 2 - Panel B needs to be described more, both in the caption and the text. For example, what do L, K, and R refer to? What is the significance of the colored bonds? Can you explain the direction of the arrows? Also, MOD still hasn’t been defined. Perhaps adding a section to the text describing these types of diagrams would be useful. 

The following explanation has been added to the caption.“A reaction mechanism is simplified as a ‘reaction rule’ in MØD. L is called Left, R is called Right, and K is called Context. The MØD algorithm looks for compounds that have fragments matching the Left and transforms the bonds between them as per the rule to produce the Right. The context can be thought of as analogous to the “intermediate step” of a conventional reaction mechanism.”

-The terminology changes from “iron-exclusive/iron-inclusive” to “Fe(+)/Fe(-)” throughout the manuscript. Please choose a consistent terminology. “Iron-exclusive” is confusing to me. One possible alternative might be “iron-containing” and “iron-absent”? Iron (-) and iron (+) are introduced around line 270. Iron-free and iron-inclusive are used in Fig 9 caption. Regardless of what is chosen, it should be consistent throughout. 

The text has been modified throughout to take this suggestion into account.

-Figure 3 - Consider labeling the top row and bottom row as “No iron” and “With iron” (or whatever terminology you decide upon). Also, this figure needs an explanation for L, K, R and the different bond/atom colors. 

The suggestion has been accepted. 

-Why does an upper mass limit of 200 units accurately capture the system? The authors present a test case with a 100 unit upper limit, but could provide a deeper justification for their choice of 200 units as an upper limit. If it’s purely a computational limit, please state this and justify why the results presented are still justified to understand these systems. 

We have added text to this effect, as well as reference to where even these low mass restrictions still predict trends in higher mass material as measured using high resolution mass spectrometry of experimental systems and various mass spectral visualization and analysis methods including Kendrick Mass Defect analysis and Van Krevelen analysis.

-A number of section in the results/discussion where many numbers are listed for the various reactions might be better presented as tables. For example, can the information in lines 163-169 be converted into a table showing reaction, # of iterations, # of products without iron, # of products with iron? 

This suggestion has been accepted.

-Figure 4 - It would really help the reader to include a legend indicating what the blue and red colors represent (rather than just having this information in the caption). The font size everywhere (tick labels, axis labels, titles) is too small (and panel A title looks to have inconsistent font size). Also, the explanation for the calculation of the predicted number of compounds for panels C and D comes after the figure is referenced in the text, which is not ideal (I think the explanation appears in line 180). 

This has been corrected.

-Figure 5 - A legend would be helpful here as well. Also, please show the actual data as points without connecting lines and the fits with dotted/dashed lines. Consider color coding the equations of fit as red or blue. Axis labels, titles, and tick labels should be larger in size. 

This suggestion has been accepted.

-Figure 6 - Is the y axis showing the number of new products with each generation or the number of unique products? The labeling is different between Figs 4/5 and 6. Is this on purpose, or are these figures showing different quantities? Please make this explicitly clear in the text and captions if so. 

This has been corrected.

-Figure 7 - Include a legend and increase font size

This has been corrected.

-Table 1 - need to label what number is being shown in the columns under “Iron -“ and “Iron +.” Also, please use consistent terminology to refer to these cases throughout the manuscript. 

This has been corrected.

-Table 2 - Why do you choose to show all the reactions for the Formose reaction in Table 1, but only show the top five reactions for the other networks studied in Table 2? Please explain why this comparison is justified. Should equivalent tables perhaps be shown for the other three reaction networks in the SI? Also, the number being shown under “Iron -“ and “Iron +” needs to be labeled. I’m also wondering why the numbers in Table 2 are so different in magnitude than those shown in Table 1 (e.g. ~10000 vs. ~10)? 

We agree with the reviewer’s suggestion. We have added the complete version of Table 2 in the SI.

The numbers are in proportion to the total rule application frequency. Formose network is denser and larger in terms of edges and nodes compared to other networks, hence the significant difference in the magnitude.

-Figure 8 - add a legend for red and blue colors; use consistent terminology in caption; increase font size everywhere. Maybe briefly describe what “degree” means in words in the caption? 

Degree has been defined at the start of section 3.4.  The plot has been corrected.

Minor comments: 

-line 9 - Corresponding author and email need to be updated

These items have been corrected.

-lines 17-18: “reaction of pyruvic acid” Into what? This phrasing isn’t parallel to the other three reactions mentioned. 

We have modified the sentence to:  the complexifying degradation reaction of pyruvic acid in water

-line 37 - does STP need to be defined?

This has been added. 

-line 37 - possibly switch “energy change” to “energy difference”?

This has been changed as suggested by the reviewer.

-line 48 - Fe^3+ - superscript is missing

This has been corrected.

-line 54 - provide a brief description of the Maillard reaction here?

A brief explanation has been provided at the end of this sentence.

-lines 60-62 - The wording of the sentence beginning with “Chemical transformations…” is complicated and difficult to parse. Perhaps rephrase as:  "Chemical transformations that are possible in the absence of iron may also be catalyzed by iron species; similarly, reactions that are impossible in the absence of iron may become possible when iron is present." 

This suggestion has been accepted.

-line 63 - “highly diversity-generating reactions” - complicated wording - consider rephrasing, perhaps as: “…includes a number of reactions that generate high chemical diversity, seeded by simple likely prebiotic reactants, including..."

This suggestion has been accepted.

-line 65 - Suggest breaking the sentence up:  "...among others. These reactions may produce hundreds to millions..."

We have accepted the reviewer’s suggestion.

-lines 70-71 - Can the authors provide another sentence of two elaborating on the graph-theory techniques and CRNRs here? This background seemed to be a little lacking throughout the introduction for the non-expert reader. 

We have added the following text: “These CRNRs are computed using reaction rules that are in silico representations of real-world chemical reaction mechanisms.

-line 74 - semicolon doesn’t seem to be grammatically correct here. Rephrase? 

We agree, this has been changed to a comma.

-line 87 - can the authors provide a rationale for choosing 25 degrees C as the lower bound on temperature? Cooler environments are occasionally suggested as plausible for the early Earth. 

We have added the following text here: “Note that while organisms have been found living at lower temperatures, there are few examples of studies of aqueous iron chemistry at lower temperatures in the literature, thus this reaction search is defined by reactions detectable over practical laboratory timescales. Presumably however, reactions that occur at higher temperatures will also proceed, albeit more slowly, at lower ones.”

-line 88 - Define Fenton reagents here?

We have added text briefly explaining this.

-line 91 - delete “of course”?

We think this sentence properly emphasizes what it intends, but to make it more clear we have deleted the comma.

-line 93-96 - Why are these two reactions specifically mentioned here? Are they outliers or difficult to classify for some reason? If so, please state this explicitly. 

These reactions are not outliers. They are mentioned here only as examples, this is now made clear in the text.

-line 126 - reaction of concentrated pyruvic acid with what? Itself? Other chemicals? The description here isn’t in parallel structure to the other reactions described. 

We have clarified this by extending the text to read “...the reaction of concentrated pyruvic acid (which degrades and self-condenses in water to give a complex product suite, e.g., [28]),...”

-line 150 - “Reactions were expanded over 1-6 iterations…” It isn’t immediately clear what this means. Can you provide a general description of this process? 

We have added the following text:”Iteration of the reactions causes the growth of networks allowing diverse products to be produced.” 

-line 155 - This seems like an odd first sentence to the results and discussion section. Perhaps start with something more general and descriptive of the overall section to help guide the reader? 

We have removed this sentence, and moved it to the start of relevant section 3.4.

-line 182 - need to specify what grows by a factor of ten - number of products, right? “e.g. the number of products in each generation grows by a factor of ten.” 

This suggestion has been accepted.

-line 184-185 - this sentence seems to be missing a word in order to make sense grammatically? 

We have rephrased this sentence per the reviewer’s suggestion.

-line 201-202 - Add a sentence comparing these results to the 200 AMU mass limit results for the five generations you followed there. It seems like the mass limit plays a huge role in the number of (new?) products - it’s ~10 for 100 AMU limit, but ~20000 for 200 AMU limit? Is this expected? 

We have added text explicating this phenomenon where these results are reported.

-line 227-228 - Where are these numbers coming from? They don’t seem to precisely match the 5th generation numbers in panel B of Figure 7

-lines 230-232 - Again, these numbers don’t seem to match what is shown in Figure 7. Also, consider turning this paragraph into a table, or alternatively just qualitatively explaining the trends (e.g. with iron, the number of reactions edges is XX-YY% more than when iron is not included). 

These numbers are total reaction rule counts, while figure 7 shows the reaction rule vs. generation distribution. We have included a table and modified the wording to avoid confusion.

-lines 240-247 - I'm confused by the overall point of this paragraph, perhaps due to the many numbers presented. Is the point that the iron-free and iron-containing networks are dominated by the non-iron containing rules? If so, what is the meaning of the increased diversity in the networks with iron present? 

The rules making up the iron-free network are present in the iron-inclusive network but their percentage share in the iron-containing network has decreased. The new-iron rules are merely creating more edges and new nodes when there is a context. The new nodes 

-lines 250-252 - These two sentences seem to contradict each other. Too many numbers. Distill down to the main point please. 

We have removed the sentence: “Moreover, the rule composition of the network remained fairly similar regardless of the presence of iron.” and attempted to simplify the text for clarity.

-line 259 - change “increment” to “increase”?

We have accepted this suggestion.

-line 271-273 - This sentence has confusing wording. Consider revising. 

We believe this sentence captures the point we are trying to make.

-line 277 - It seems like the pyruvic acid and Formose reactions are quite different in % catalyzed than the other two reactions. Can you offer speculation as to why this might be?

We have added the following text: “The percentage catalysis for the Glucose Degradation Reaction and Maillard Reaction are lesser compared to the other two networks because of different numbers of generations. Furthermore, different reactants pose different reaction spheres leading to differences in catalysis intensity.” 

-lines 302-304 - That’s only for pyruvic acid and Formose reaction, though, right? They also appear fairly high in glucose degradation, but not at all in the Maillard reaction. Why might this be? 

We have added the following text: “This effect can also be seen in the iron-containing graph where Fe(II) and Fe(III) assume the third and fourth place positions in terms of node degree in the formose and the pyruvic network. They also appear at fifth and sixth positions in the Glucose Degradation network, and at sixteenth and nineteenth positions in the Maillard Reaction Network (Refer SI).”

-line 312 - define Gephi

The definition has been added and a reference to the software is included.

-line 316 - remove “below” 

This has been corrected.

-line 319-320 - This wasn’t immediately obvious to me when first looking at the plots. Can you offer the reader a bit more guidance as to what they are looking for/at? 

We have added explanatory text below the figure which helps explain this.

-line 324 - change “CNRNs” to “CRNRs”

This has been corrected.

Lines 333-336 - Is this the key point of the study? If so, I think it could be made a bit more explicitly throughout the results/discussion section. Perhaps because of the many numbers listed, it’s hard for the reader to parse the information and follow the main conclusions? Please also make sure this statement is consistent with the last sentence of the abstract. 

This has been corrected.

-line 338 - The conclusion section uses the words “in silico” a fair amount, when this has only been sparsely mentioned before. Can you make this more consistent throughout the paper? 

We have attempted to make this usage more uniform throughout.

-line 338-339: “… metal, specifically iron, catalysis…” wording is a little odd. Perhaps revise to: “…how the addition of metal catalysis (specifically iron) can modify the evolution of CRNRs.”

We have accepted this suggestion. 

-line 341-343 - So what guarantees do you have that your conclusions here are representative of the general case with additional reactions/rules? 

We have modified the text to read “While this database is extensive, it is likely that data mining under-represents the extent of iron-catalyzed small molecule catalysis, as there likely remain many undiscovered and unreported iron-involving reaction mechanisms.”

-line 344 - how might pH, temperature, and concentration potentially affect these results? What uncertainties do they imply for your conclusions here? 

The text has been modified to read, “Furthermore, this type of modeling does not take into account the effects of pH, temperature or concentration on the relative rates of reactions, all of which can significantly skew relative product concentrations, it merely presents a road-map of possible reactions. As such, such models offer good first-order ways to explore potential CRN product diversity, which can be useful for experimental studies.”

-line 346 - The sentence starting with “Nevertheless” is very long. Consider breaking it into multiple sentences. 

This sentence has been broken into shorter sentences as suggested by the reviewer.

-line 348-349 - Regarding the statement: “…these CRNRs can be queried to find autocatalytic reaction loops and reaction sequences which cycle iron between its +2 and +3 redox state…” - this is the first mention of this as far as I can tell. Was this done at all in your study? Or could it be invoked in the future? How would this be done?

We have provided a reference to the methodology used previously for this purpose, and added a statement that this follow-on work is planned.

-line 352 - reference to organic geochemistry here felt like it was just thrown in at the end, without much actual motivation for it throughout the paper.

We think the reviewer is referring to inorganic geochemistry, we have attempted to introduce this concept in the introduction, and tie it back together in the conclusion.

Round 2

Reviewer 1 Report

The authors have given a reasonable response to my earlier questions and I now recommend acceptance.

Reviewer 2 Report

I appreciate the authors taking the time to make these revisions. I think the manuscript has been substantially improved and is acceptable for publication in its present form.